# Training Generative Adversarial Networks via Primal-Dual Subgradient Methods: A Lagrangian Perspective on GAN

[†]**Xu Chen** , [‡]**Jiang Wang,** [†]**Hao Ge** [*]
[†] Department of EECS, Northwestern University, Evanston, IL, USA
[‡] Google Inc.
{chenx,haoge2013}@u.northwestern.edu
wangjiangb@gmail.com

## Abstract

We relate the minimax game of generative adversarial networks (GANs) to finding the saddle points of the Lagrangian function for a convex optimization problem, where the discriminator outputs and the distribution of generator outputs play the roles of primal variables and dual variables, respectively. This formulation shows the connection between the standard GAN training process and the primal-dual subgradient methods for convex optimization. The inherent connection does not only provide a theoretical convergence proof for training GANs in the function space, but also inspires a novel objective function for training. The modified objective function forces the distribution of generator outputs to be updated along the direction according to the primal-dual subgradient methods. A toy example shows that the proposed method is able to resolve mode collapse, which in this case cannot be avoided by the standard GAN or Wasserstein GAN. Experiments on both Gaussian mixture synthetic data and real-world image datasets demonstrate the performance of the proposed method on generating diverse samples.

## 1 Introduction

Generative adversarial networks (GANs) are a class of game theoretical methods for learning data distributions. It trains the generative model by maintaining two deep neural networks, namely the discriminator network $D$ and the generator network $G$. The generator aims to produce samples resembling real data samples, while the discriminator aims to distinguish the generated samples and real data samples.

The standard GAN training procedure is formulated as the following minimax game:

$$\min_G \max_D \mathsf{E}_{\boldsymbol{x} \sim p_d(\boldsymbol{x})}\{\log D(\boldsymbol{x})\} + \mathsf{E}_{\boldsymbol{z} \sim p_z(\boldsymbol{z})}\{\log(1 - D(G(\boldsymbol{z})))\}, \tag{1}$$

where $p_d(\boldsymbol{x})$ is the data distribution and $p_z(\boldsymbol{z})$ is the noise distribution. The generated samples $G(\boldsymbol{z})$ induces a generated distribution $p_g(\boldsymbol{x})$. Theoretically, the optimal solution to (1) is $p_g^* = p_d$ and $D^*(\boldsymbol{x}) = 1/2$ for all $\boldsymbol{x}$ in the support of data distribution.

In practice, the discriminator network and the generator network are parameterized by $\boldsymbol{\theta}_d$ and $\boldsymbol{\theta}_g$, respectively. The neural network parameters are updated iteratively according to gradient descent. In particular, the discriminator is first updated either with multiple gradient descent steps until convergence or with a single gradient descent step, then the generator is updated with a single descent step. However, the analysis of the convergence properties on the training approaches is challenging, as noted by Ian Goodfellow in (Goodfellow, 2016), "For GANs, there is no theoretical prediction as to whether simultaneous gradient descent should converge or not. Settling this theoretical question, and developing algorithms guaranteed to converge, remain important open research problems.". There have been some recent studies on the convergence behaviours of GAN training (Nowozin et al., 2016; Li et al., 2017b; Heusel et al., 2017; Nagarajan & Kolter, 2017; Mescheder et al., 2017).

---

[*]The first two authors have equal contributions.

The simultaneous gradient descent method is proved to converge assuming the objective function is convex-concave in the network parameters (Nowozin et al., 2016). The local stability property is established in (Heusel et al., 2017; Nagarajan & Kolter, 2017).

One notable inconvergence issue with GAN training is referred to as mode collapse, where the generator characterizes only a few modes of the true data distribution (Goodfellow et al., 2014; Li et al., 2017b). Various methods have been proposed to alleviate the mode collapse problem. Feature matching for intermediate layers of the discriminator has been proposed in (Salimans et al., 2016). In (Metz et al., 2016), the generator is updated based on a sequence of previous unrolled discriminators. A mixture of neural networks are used to generate diverse samples (Tolstikhin et al., 2017; Hoang et al., 2017; Arora et al., 2017). In (Arjovsky & Bottou, 2017), it was proposed that adding noise perturbation on the inputs to the discriminator can alleviate the mode collapse problem. It is shown that this training-with-noise technique is equivalent to adding a regularizer on the gradient norm of the discriminator (Roth et al., 2017). The Wasserstein divergence is proposed to resolve the problem of incontinuous divergence when the generated distribution and the data distribution have disjoint supports (Arjovsky et al., 2017; Gulrajani et al., 2017). Mode regularization is used in the loss function to penalize the missing modes (Che et al., 2016; Srivastava et al., 2017). The regularization is usually based on heuristics, which tries to minimize the distance between the data samples and the generated samples, but lacks theoretical convergence guarantee.

In this paper, we formulate the minimax optimization for GAN training (1) as finding the saddle points of the Lagrangian function for a convex optimization problem. In the convex optimization problem, the discriminator function $D(\cdot)$ and the probabilities of generator outputs $p_g(\cdot)$ play the roles of the primal variables and dual variables, respectively. This connection not only provides important insights in understanding the convergence of GAN training, but also enables us to leverage the primal-dual subgradient methods to design a novel objective function that helps to alleviate mode collapse. A toy example reveals that for some cases when standard GAN or WGAN inevitably leads to mode collapse, our proposed method can effectively avoid mode collapse and converge to the optimal point.

In this paper, we do not aim at achieving superior performance over other GANs, but rather provide a new perspective of understanding GANs, and propose an improved training technique that can be applied on top of existing GANs. The contributions of the paper are as follows:

- The standard training of GANs in the function space is formulated as primal-dual subgradient methods for solving convex optimizations.

- This formulation enables us to show that with a proper gradient descent step size, updating the discriminator and generator probabilities according to the primal-dual algorithms will provably converge to the optimal point.

- This formulation results in a novel training objective for the generator. With the proposed objective function, the generator is updated such that the probabilities of generator outputs are pushed to the optimal update direction derived by the primal-dual algorithms. Experiments have shown that this simple objective function can effectively alleviate mode collapse in GAN training.

- The convex optimization framework incorporates different variants of GANs including the family of $f$-GAN (Nowozin et al., 2016) and an approximate variant of WGAN. For all these variants, the training objective can be improved by including the optimal update direction of the generated probabilities.

## 2 PRIMAL-DUAL SUBGRADIENT METHODS FOR CONVEX OPTIMIZATION

In this section, we first describe the primal-dual subgradient methods for convex optimization. Later, we explicitly construct a convex optimization and relate the subgradient methods to standard GAN training. Consider the following convex optimization problem:

$$\text{maximize } f_0(\boldsymbol{x}) \tag{2a}$$
$$\text{subject to } f_i(\boldsymbol{x}) \geq 0, i = 1, \cdots, \ell \tag{2b}$$
$$\boldsymbol{x} \in X, \tag{2c}$$

where $\boldsymbol{x} \in \mathbb{R}^k$ is a length-$k$ vector, $X$ is a convex set, and $f_i(\boldsymbol{x})$, $i = 0 \cdots, \ell$, are concave functions mapping from $\mathbb{R}^k$ to $\mathbb{R}$. The Lagrangian function is calculated as

$$L(\boldsymbol{x}, \boldsymbol{\lambda}) = f_0(\boldsymbol{x}) + \sum_{i=1}^{\ell} \lambda_i f_i(\boldsymbol{x}). \tag{3}$$

In the optimization problem, the variables $\boldsymbol{x} \in \mathbb{R}^k$ and $\boldsymbol{\lambda} \in \mathbb{R}_+^{\ell}$ are referred to as primal variables and dual variables, respectively. The primal-dual pair $(\boldsymbol{x}^*, \boldsymbol{\lambda}^*)$ is a saddle-point of the Lagrangian fuction, if it satisfies:

$$L(\boldsymbol{x}^*, \boldsymbol{\lambda}^*) = \min_{\boldsymbol{\lambda} \geq 0} \max_{\boldsymbol{x} \in X} L(\boldsymbol{x}, \boldsymbol{\lambda}). \tag{4}$$

Primal-dual subgradient methods have been widely used to solve the convex optimization problems, where the primal and dual variables are updated iteratively, and converge to a saddle point (Nedić & Ozdaglar, 2009; Komodakis & Pesquet, 2015).

There are two forms of algorithms, namely dual-driven algorithm and primal-dual-driven algorithm. For both approaches, the dual variables are updated according to the subgradient of $L(\boldsymbol{x}(t), \boldsymbol{\lambda}(t))$ with respect to $\boldsymbol{\lambda}(t)$ at each iteration $t$. For the dual-driven algorithm, the primal variables are updated to achieve maximum of $L(\boldsymbol{x}, \boldsymbol{\lambda}(t))$ over $\boldsymbol{x}$. For the primal-dual-driven algorithm, the primal variables are updated according to the subgradient of $L(\boldsymbol{x}(t), \boldsymbol{\lambda}(t))$ with respect to $\boldsymbol{x}(t)$. The iterative update process is summarized as follows:

$$\boldsymbol{x}(t+1) = \begin{cases} \arg\max_{\boldsymbol{x} \in X} L(\boldsymbol{x}, \boldsymbol{\lambda}(t)) & \text{(dual-driven algorithm)} \\ \mathcal{P}_X [\boldsymbol{x}(t) + \alpha(t) \partial_{\boldsymbol{x}} L(\boldsymbol{x}(t), \boldsymbol{\lambda}(t))] & \text{(primal-dual-driven algorithm)} \end{cases} \tag{5}$$

$$\boldsymbol{\lambda}(t+1) = [\boldsymbol{\lambda}(t) - \alpha(t) \partial_{\boldsymbol{\lambda}} L(\boldsymbol{x}(t), \boldsymbol{\lambda}(t))]_+, \tag{6}$$

where $\mathcal{P}_X(\cdot)$ denotes the projection on set $X$ and $(x)_+ = \max(x, 0)$.

The following theorem proves that the primal-dual subgradient methods will make the primal and dual variables converge to the optimal solution of the convex optimization problem.

**Theorem 1** *Consider the convex optimization* (2). *Assume the set of saddle points is compact. Suppose $f_0(\boldsymbol{x})$ is a strictly concave function over $\boldsymbol{x} \in X$ and the subgradient at each step is bounded. There exists some step size $\alpha^{(t)}$ such that both the dual-driven algorithm and the primal-dual-driven algorithm yield $\boldsymbol{x}^{(t)} \to \boldsymbol{x}^*$ and $\boldsymbol{\lambda}^{(t)} \to \boldsymbol{\lambda}^*$, where $\boldsymbol{x}^*$ is the solution to* (2), *and $\boldsymbol{\lambda}^*$ satisfies*

$$L(\boldsymbol{x}^*, \boldsymbol{\lambda}^*) = \max_{\boldsymbol{x} \in X} L(\boldsymbol{x}, \boldsymbol{\lambda}^*). \tag{7}$$

**Proof:** See Appendix 7.1.

## 3 Training GAN via Primal-Dual Subgradient Methods

### 3.1 GAN as a convex optimization

We explicitly construct a convex optimization problem and relate it to the minimax game of GANs. We assume that the source data and generated samples belong to a finite set $\{\boldsymbol{x}_1, \cdots, \boldsymbol{x}_n\}$ of arbitrary size $n$. The extension to uncountable sets can be derived in a similar manner (Luenberger, 1997). The finite case is of particular interest, because any real-world data has a finite size, albeit the size could be arbitrarily large.

We construct the following convex optimization problem:

$$\text{maximize} \sum_{i=1}^{n} p_d(\boldsymbol{x}_i) \log(D_i) \tag{8a}$$

$$\text{subject to } \log(1 - D_i) \geq \log(1/2), i = 1, \cdots, n \tag{8b}$$

$$\boldsymbol{D} \in \mathcal{D}, \tag{8c}$$

where $\mathcal{D}$ is some convex set. The primal variables are $\boldsymbol{D} = (D_1, \cdots, D_n)$, where $D_i$ is defined as $D_i = D(\boldsymbol{x}_i)$. Let $\boldsymbol{p}_g = (p_g(\boldsymbol{x}_1), \cdots, p_g(\boldsymbol{x}_n))$, where $p_g(\boldsymbol{x}_i)$ is the Lagrangian dual associated with the $i$-th constraint. The Lagrangian function is thus

$$L(\boldsymbol{D}, \boldsymbol{p}_g) = \sum_{i=1}^n p_d(\boldsymbol{x}_i) \log(D_i) + \sum_{i=1}^n p_g(\boldsymbol{x}_i) \log(2(1 - D_i)), \boldsymbol{D} \in \mathcal{D}. \tag{9}$$

When $\mathcal{D} = \{\boldsymbol{D} : 0 \leq D_i \leq 1, \forall i\}$, finding the saddle points for the Lagrangian function is exactly equivalent to solving the GAN minimax problem(1). This inherent connection enables us to utilize the primal-dual subgradient methods to design update rules for $D(\boldsymbol{x})$ and $p_g(\boldsymbol{x})$ such that they converge to the saddle points. The following theorem provides a theoretical guideline for the training of GANs.

**Theorem 2** *Consider the Lagrangian function given by* (9) *with* $\mathcal{D} = \{\boldsymbol{D} : \epsilon \leq D_i \leq 1 - \epsilon, \forall i\}$, *where* $0 < \epsilon < 1/2$. *If the discriminator and generator have enough capacity, and the discriminator output and the generated distribution are updated according to the primal-dual update rules* (5) *and* (6) *with* $(\boldsymbol{x}, \boldsymbol{\lambda}) = (\boldsymbol{D}, \boldsymbol{p}_g)$, *then* $p_g(\cdot)$ *converges to* $p_d(\cdot)$.

**Proof:** The optimization problem (8) is a particularized form of (2), where $f_0(\cdot) = \sum_{i=1}^n p_d(\boldsymbol{x}_i) \log(D_i)$, $f_i(\cdot) = \log(1 - D_i)$ and $X = [\epsilon, 1 - \epsilon]^n$. The objective function is strictly concave over $\boldsymbol{D}$. Moreover, since $\boldsymbol{D}$ is projected onto the compact set $[\epsilon, 1 - \epsilon]$ at each iteration $t$, the subgradients $\partial f_i(\boldsymbol{D}^{(t)})$ are bounded. The assumptions of Theorem 1 are satisfied.

Since the constraint (8b) gives an upper bound of $D_i \leq 1/2$, the solution to the above convex optimization is obviously $D_i^* = 1/2$, for all $i = 1, \cdots, n$. Since the problem is convex, the optimal primal solution is the primal saddle point of the Lagrangian function (Bertsekas, 1999, Chapter 5). Moreover, any primal-dual saddle point $(\boldsymbol{D}^*, \boldsymbol{p}_g^*)$ satisfies $L(\boldsymbol{D}^*, \boldsymbol{p}_g^*) = \max_{\boldsymbol{D} \in \mathcal{D}} L(\boldsymbol{D}, \boldsymbol{p}_g^*)$. Since $\boldsymbol{D}^*$ is strictly inside $\mathcal{D}$, we have $\partial_{\boldsymbol{D}} L(\boldsymbol{D}^*, \boldsymbol{p}_g^*) = 0$. Since $\partial_{D_i} L(\boldsymbol{D}^*, \boldsymbol{p}_g^*) = 2p_d(\boldsymbol{x}_i) - 2p_g^*(\boldsymbol{x}_i)$, we have $p_g^* = p_d$, and the saddle point is unique. By Theorem 1, the primal-dual update rules will guarantee convergence of $\left(\boldsymbol{D}^{(t)}, \boldsymbol{p}_g^{(t)}\right)$ to the primal-dual saddle point $(\boldsymbol{D}^*, \boldsymbol{p}_g^*)$. $\qquad\square$

It can be seen that the standard training of GAN corresponds to either dual-driven algorithm (Nowozin et al., 2016) or primal-dual-driven algorithm (Arjovsky et al., 2017; Goodfellow et al., 2014). A natural question arises: Why does the standard training fail to converge and lead to mode collapse? As will be shown later, the underlying reason is that standard training of GANs in some cases do not update the generated distribution according to (6). Theorem 2 inspires us to propose a training algorithm to tackle this issue.

### 3.2 ALGORITHM DESCRIPTION

First, we present our training algorithm. Later, we will use a toy example to give intuitions of why our algorithm is effective to avoid mode collapse.

The algorithm is described in Algorithm 1. The maximum step of discriminator update is $k_0$. In the context of primal-dual-driven algorithms, $k_0 = 1$. In the context of dual-driven algorithms, $k_0$ is some large constant, such that the discriminator is updated till convergence at each training epoch. The update of the discriminator is the same as standard GAN training. The main difference is the modified loss function for the generator update (13). The intuition is that when the generated samples have disjoint support from the data, the generated distribution at the data support may not be updated using standard training. This is exactly one source of mode collapse. Ideally, the modified loss function will always update the generated probabilities at the data support along the optimal direction.

The generated probability mass at $\boldsymbol{x}$ is $p_g(\boldsymbol{x}) = \frac{1}{m} \sum_{i=1}^m 1\{G(\boldsymbol{z}_i) = \boldsymbol{x}\}$, where $1\{\cdot\}$ is the indicator function. The indicator function is not differentiable, so we use a continuous kernel to approximate it. Define

$$k_\sigma(\boldsymbol{x}) = e^{-\frac{||\boldsymbol{x}||^2}{\sigma^2}}, \tag{14}$$

where $\sigma$ is some positive constant. The constant $\sigma$ is also called bandwidth for kernel density estimation. The empirical generated distribution is thus approximately calculated as (17). There

---

**Algorithm 1** Training GAN via Primal-Dual Subgradient Methods

---

**Initialization**: Choose the objective function $f_0(\cdot)$ and constraint function $f_1(\cdot)$ according to the GAN realization. For the original GAN based on Jensen-Shannon divergence, $f_0(D) = \log(D)$ and $f_1(D) = \log(2(1 - D))$.

**while** the stopping criterion is not met **do**

    Sample minibatch $m_1$ data samples $\boldsymbol{x}_1, \cdots, \boldsymbol{x}_{m_1}$.

    Sample minibatch $m_2$ noise samples $\boldsymbol{z}_1, \cdots, \boldsymbol{z}_{m_2}$.

    **for** $k = 1, \cdots, k_0$ **do**

        Update the discriminator parameters with gradient ascent:

$$\nabla_{\boldsymbol{\theta}_d} \left[ \frac{1}{m_1} \sum_{i=1}^{m_1} f_0(D(\boldsymbol{x}_i)) + \frac{1}{m_2} \sum_{j=1}^{m_2} f_1\left(D\left(G\left(\boldsymbol{z}_j\right)\right)\right) \right]. \tag{10}$$

    **end for**

    Update the target generated distribution as:

$$\tilde{p}_g(\boldsymbol{x}_i) = p_g(\boldsymbol{x}_i) - \alpha f_1(D(\boldsymbol{x}_i)), i = 1, \cdots, m_1, \tag{11}$$

    where $\alpha$ is some step size and

$$p_g(\boldsymbol{x}_i) = \frac{1}{m_2} \sum_{j=1}^{m_2} k_\sigma(G(\boldsymbol{z}_j) - \boldsymbol{x}_i). \tag{12}$$

    With $\tilde{p}_g(\boldsymbol{x}_i)$ *fixed*, update the generator parameters with gradient descent:

$$\nabla_{\boldsymbol{\theta}_g} \left[ \frac{1}{m_2} \sum_{j=1}^{m_2} f_1\left(D\left(G\left(\boldsymbol{z}_j\right)\right)\right) + \frac{1}{m_1} \sum_{i=1}^{m_1} \left( \tilde{p}_g(\boldsymbol{x}_i) - \frac{1}{m_2} \sum_{j=1}^{m_2} k_\sigma(G(\boldsymbol{z}_j) - \boldsymbol{x}_i) \right)^2 \right]. \tag{13}$$

**end while**

---

are different bandwidth selection methods (Botev et al., 2010; Hall et al., 1991). It can be seen that as $\sigma \to 0$, $k_\sigma(\boldsymbol{x} - \boldsymbol{y})$ tends to the indicator function, but it will not give large enough gradients to far areas that experience mode collapse. A larger $\sigma$ implies a coarser quantization of the space in approximating the distribution. In practical training, the kernel bandwidth can be set larger at first and gradually decreases as the iteration continues.

By the dual update rule (6), the generated probability of every $\boldsymbol{x}_i$ should be updated as

$$\tilde{p}_g(\boldsymbol{x}_i) = p_g(\boldsymbol{x}_i) - \alpha \frac{\partial L(\boldsymbol{D}, \boldsymbol{p}_g)}{\partial p_g(\boldsymbol{x}_i)} \tag{15}$$

$$= p_g(\boldsymbol{x}_i) - \alpha \log(2(1 - D(\boldsymbol{x}_i))). \tag{16}$$

This motivates us to add the second term of (13) in the loss function, such that the generated distribution is pushed towards the target distribution (15).

Although having good convergence guarantee in theory, the non-parametric kernel density estimation of the generated distribution may suffer from the curse of dimension. Previous works combining kernel learning and the GAN framework have proposed methods to scale the algorithms to deal with high-dimensional data, and the performances are promising (Li et al., 2015; 2017a; Sinn & Rawat, 2017). One common method is to project the data onto a low dimensional space using an autoencoder or a bottleneck layer of a pretrained neural network, and then apply the kernel-based estimates on the feature space. Using this approach, the estimated probability of $\boldsymbol{x}_i$ becomes

$$p_g(\boldsymbol{x}_i) = \frac{1}{m_2} \sum_{j=1}^{m_2} k_\sigma(f_\phi(G(\boldsymbol{z}_j)) - f_\phi(\boldsymbol{x}_i)), \tag{17}$$

where $f_\phi(.)$ is the projection of the data to a low dimensional space. We will leave the work of generating high-resolution images using this approach as future work.

### 3.3 Intuition of avoiding mode collapse

Mode collapse occurs when the generated samples have a very small probability to overlap with some families of the data samples, and the discriminator $D(\cdot)$ is locally constant around the region of the generated samples. We use a toy example to show that the standard training of GAN and Wasserstein may fail to avoid mode collapse, while our proposed method can succeed.

**Claim 1** *Suppose the data distribution is $p_d(x) = 1\{x = 1\}$, and the initial generated distribution is $p_g(x) = 1\{x = 0\}$. The discriminator output $D(x)$ is some function that is equal to zero for $|x - 0| \leq \delta$ and is equal to one for $|x - 1| \leq \delta$, where $0 < \delta < 1/2$. Standard training of GAN and WGAN leads to mode collapse.*

**Proof:** We first show that the discriminator is not updated, and then show that the generator is not updated during the standard training process.

In standard training of GAN and WGAN, the discriminator is updated according to the gradient of (10). For GAN, since $0 \leq D(x) \leq 1$, the objective funtion for the discriminator is at most zero, i.e.,

$$\mathsf{E}_{p_d} \log\left(D\left(\boldsymbol{x}\right)\right) + \mathsf{E}_{p_g} \log\left(1 - D\left(\boldsymbol{x}\right)\right) = \log(D(1)) + \log(1 - D(0)) \leq 0, \tag{18}$$

which is achieved by the current $D(x)$ by assumption.

For WGAN, the optimal discrminator output $D(x)$ is some 1-Lipschitz function such that $\mathsf{E}_{p_d}\{D(x)\} - \mathsf{E}_{p_g}\{D(x)\}$ is maximized. Since

$$\mathsf{E}_{p_d}\{D(x)\} - \mathsf{E}_{p_g}\{D(x)\} = D(1) - D(0) \leq 1, \tag{19}$$

where (19) is due to the Lipschitz condition $|D(1) - D(0)| \leq 1$. The current $D(x)$ is obviously optimal. Thus, for both GAN and WGAN, the gradient of the loss function with respect to $\boldsymbol{\theta}_d$ is zero and the discriminator parameters are not updated.

On the other hand, in standard training, the generator parameters $\boldsymbol{\theta}_g$ are updated with only the first term of (13). By the chain rule,

$$\partial_{\boldsymbol{\theta}_g} \log\left(1 - D\left(G\left(\boldsymbol{z}_i\right)\right)\right) = -\frac{1}{1 - D\left(G\left(\boldsymbol{z}_i\right)\right)} \partial_x D(x)\big|_{x=G(\boldsymbol{z}_i)} \partial_{\boldsymbol{\theta}_g} G(\boldsymbol{z}_i) \tag{20}$$

$$= 0, \tag{21}$$

where (21) is due to the assumption that $D(x)$ is locally constant for $x = 0$. Therefore, the generator and the discriminator reach a local optimum point. The generated samples are all zeros. $\square$

In our proposed training method, when $x = 1$, the optimal update direction is given by (11), where $\tilde{p}_g$ is a large value because $D(1) = 1$. Therefore, by (13), the second term in the loss function is very large, which forces the generator to generate samples at $G(\boldsymbol{z}) = 1$. As the iteration continues, the generated distribution gradually converges to data distribution, and $D(x)$ gradually converges to $1/2$, which makes $\partial_{p_g(x)} L(D(x), p_g(x)) = \log(2(1 - D(x)))$ become zero. The experiment in Section 5 demonstrates this training dynamic.

In this paper, the standard training of GANs in function space has been formulated as primal-dual updates for convex optimization. However, the training is optimized over the network parameters in practice, which typically yields a non-convex non-concave problem. Theorem 2 tells us that as long as the discriminator output and the generated distribution are updated according to the primal-dual update rule, mode collapse should not occur. This insight leads to the addition of the second term in the modified loss function for the generator (13). In Section 5, experiments on the above-mentioned toy example and real-world datasets show that the proposed training technique can greatly improve the baseline performance.

## 4 Variants of GANs

Consider the following optimization problem:

$$\text{maximize} \sum_{i=1}^{n} p_d(\boldsymbol{x}_i) f_0(D_i) \tag{22a}$$

$$\text{subject to } f_1(D_i) \geq 0, i = 1, \cdots, n, \tag{22b}$$

Table 1: Variants of GANs under the convex optimization framework.

| Divergence metric | $f_0(D_i)$ | $f_1(D_i)$ | $D_i^*$ |
|---|---|---|---|
| Kullback-Leibler | $\log(D_i)$ | $1 - D_i$ | $\frac{p_d(\boldsymbol{x}_i)}{p_g(\boldsymbol{x}_i)}$ |
| Reverse KL | $-D_i$ | $\log D_i$ | $\frac{p_g(\boldsymbol{x}_i)}{p_d(\boldsymbol{x}_i)}$ |
| Pearson $\chi^2$ | $D_i$ | $-\frac{1}{4}D_i^2 - D_i$ | $\frac{2(p_d(\boldsymbol{x}_i) - p_g(\boldsymbol{x}_i))}{p_g(\boldsymbol{x}_i)}$ |
| Squared Hellinger $\chi^2$ | $1 - D_i$ | $1 - 1/D_i$ | $\sqrt{\frac{p_g(\boldsymbol{x}_i)}{p_d(\boldsymbol{x}_i)}}$ |
| Jensen-Shannon | $\log(D_i)$ | $\log(1 - D_i) - \log(1/2)$ | $\frac{p_d(\boldsymbol{x}_i)}{p_d(\boldsymbol{x}_i) + p_g(\boldsymbol{x}_i)}$ |
| Approximate WGAN | $D_i - \epsilon D_i^2$ | $-D_i$ | $\frac{p_d(\boldsymbol{x}_i) - p_g(\boldsymbol{x}_i)}{2\epsilon p_d(\boldsymbol{x}_i)}$ |
| Other metric | $-\frac{1}{2}D_i^2 + D_i$ | $D_i - 2$ | $\frac{p_d(\boldsymbol{x}_i) + p_g(\boldsymbol{x}_i)}{p_d(\boldsymbol{x}_i)}$ |

where $f_0(\cdot)$ and $f_1(\cdot)$ are concave functions. Compared with the generic convex optimization problem (2), the number of constraint functions is set to be the variable alphabet size, and the constraint functions are $f_i(\boldsymbol{D}) = f_1(D_i)$, $i = 1, \cdots, n$.

The objective and constraint functions in (22) can be tailored to produce different GAN variants. For example, Table 1 shows the large family of $f$-GAN (Nowozin et al., 2016). The last row of Table 1 gives a new realization of GAN with a unique saddle point of $D^*(x) = 2$ and $p_g(\boldsymbol{x}) = p_d(\boldsymbol{x})$.

We also derive a GAN variant similar to WGAN, which is named "Approximate WGAN". As shown in Table 1, the objective and constraint functions yield the following minimax problem:

$$\min_G \max_D \mathsf{E}_{\boldsymbol{x} \sim p_d(\boldsymbol{x})} \left\{ D(\boldsymbol{x}) - \epsilon D^2(\boldsymbol{x}) \right\} - \mathsf{E}_{\boldsymbol{x} \sim p_g(\boldsymbol{x})} \left\{ D(\boldsymbol{x}) \right\}, \qquad (23)$$

where $\epsilon$ is an arbitrary positive constant. The augmented term $\epsilon D^2(\boldsymbol{x})$ is to make the objective function strictly concave, without changing the original solution. It can be seen that this problem has a unique saddle point $p_g^*(\boldsymbol{x}) = p_d(\boldsymbol{x})$. As $\epsilon$ tends to 0, the training objective function becomes identical to WGAN. The optimal $D(\boldsymbol{x})$ for WGAN is some Lipschitz function that maximizes $\mathsf{E}_{\boldsymbol{x} \sim p_d(\boldsymbol{x})} \{D(\boldsymbol{x})\} - \mathsf{E}_{\boldsymbol{x} \sim p_g(\boldsymbol{x})} \{D(\boldsymbol{x})\}$, while for our problem is $D^*(\boldsymbol{x}) = 0$. Weight clipping can still be applied, but serves as a regularizer to make the training more robust (Merolla et al., 2016).

The training algorithms for these variants of GANs follow by simply changing the objective function $f_0(\cdot)$ and constraint function $f_1(\cdot)$ accordingly in Algorithm 1.

## 5 EXPERIMENTS

### 5.1 SYNTHETIC DATA

Fig. 1 shows the training performance for a toy example. The data distribution is $p_g(x) = 1\{x = 1\}$. The inital generated samples are concentrated around $x = -3.0$. The details of the neural network parameters can be seen in Appendix 7.3. Fig. 1a shows the generated samples in the 90 quantile as the training iterates. After 8000 iterations, the generated samples from standard training of GAN and WGAN are still concentrated around $x = -3.0$. As shown in Fig. 1c and 1d, the discrminators hardly have any updates throughout the training process. Using the proposed training approach, the generated samples gradually converge to the data distribution and the discriminator output converges to the optimal solution with $D(1) = 1/2$.

Fig. 2 shows the performance of the proposed method for a mixture of 8 Gaussain data on a circle. While the original GANs experience mode collapse (Nguyen et al., 2017; Metz et al., 2016), our proposed method is able to generate samples over all 8 modes. In the training process, the bandwidth of the Gaussian kernel (14) is inialized to be $\sigma^2 = 0.1$ and decreases at a rate of $0.8^{\frac{t}{2000}}$, where $t$ is the iteration number. The generated samples are dispersed initially, and then gradually converge to the Gaussian data samples. Note that our proposed method involves a low complexity with a simple regularization term added in the loss function for the generator update.

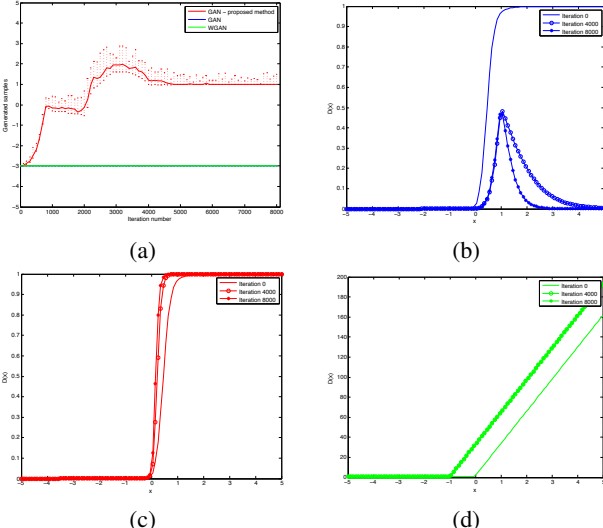

Figure 1: Performance of a toy example. Figure (a) shows the generated samples for different GANs. Figure (b) shows the discriminator output $D(x)$ for GANs trained using the proposed method. Figure (c) and (d) show the discriminator output $D(x)$ for standard GANs and WGANs.

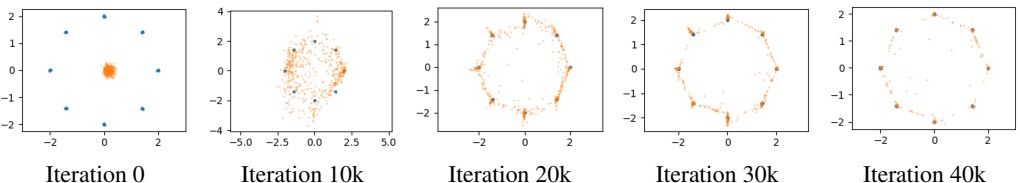

| Iteration 0 | Iteration 10k | Iteration 20k | Iteration 30k | Iteration 40k |

Figure 2: Performance of the proposed algorithm on 2D mixture of Gaussian data. The data samples are marked in blue and the generated samples are marked in orange.

## 5.2 REAL-WORLD DATASETS

We also evaluate the performance of the proposed method on two real-world datasets: MNIST and CIFAR-10. Please refer to the appendix for detailed architectures. *Inception score* (Salimans et al., 2016) is employed to evaluate the proposed method. It applies a pretrained inception model to every generated image to get the conditional label distribution $p(y|\mathbf{x})$. The Inception score is calculated as $\exp\left(\mathsf{E}_{\boldsymbol{x}}\left\{\mathrm{KL}(p(y|x) \parallel p(y))\right\}\right)$. It measures the quality and diversity of the generated images.

### 5.2.1 MNIST

The MNIST dataset contains 60000 labeled images of $28 \times 28$ grayscale digits. We train a simple LeNet-5 convolutional neural network classifier on MNIST dataset that achieves 98.9% test accuracy, and use it to compute the inception score. The proposed method achieves an inception score of 9.8, while the baseline method achieves an inception score of 8.8. The examples of generated images are shown in Fig. 3. The generated images are almost indistinguishable from real images.

We further evaluated our algorithm on an augmented 1000-class MNIST dataset to further demonstrate the robustness of the proposed algorithm against mode collapse problem. More details of the experimental results can be found in the Appendix.

### 5.2.2 CIFAR-10

CIFAR is a natural scene dataset of $32 \times 32$. We use this dataset to evaluate the visual quality of the generated samples. Table 2 shows the inception scores of different GAN models on CIFAR-10 dataset. The inception score of the proposed model is much better than the baseline method WGAN

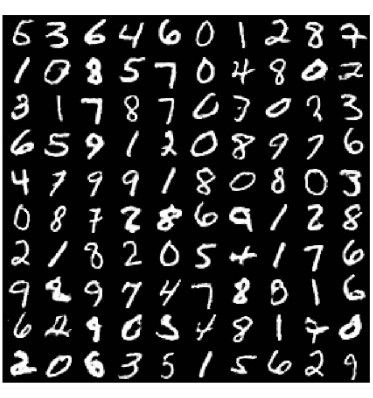 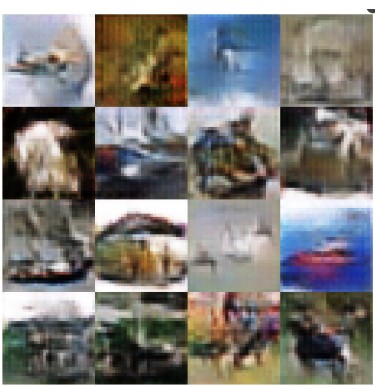

MNIST                                    CIFAR

Figure 3: Examples of generated images using MNIST and CIFAR dataset.

| Method | Score |
|---|---|
| Real data | $11.24 \pm 0.16$ |
| WGAN (Arjovsky et al., 2017) | $3.82 \pm 0.06$ |
| MIX + WGAN (Arora et al., 2017) | $4.04 \pm 0.07$ |
| Improved-GAN (Salimans et al., 2016) | $4.36 \pm 0.04$ |
| ALI (Dumoulin et al., 2016) | $5.34 \pm 0.05$ |
| DCGAN (Radford et al., 2015) | $6.40 \pm 0.05$ |
| Proposed method | $4.53 \pm 0.04$ |

Table 2: Inception scores on CIFAR-10 dataset.

that uses similar network architecture and training method. Note that although DCGGAN achieves a better score, it uses a more complex network architecture. Examples of the generated images are shown in Fig. 3.

## 6  CONCLUSION

In this paper, we propose a primal-dual formulation for generative adversarial learning. This formulation interprets GANs from the perspective of convex optimization, and gives the optimal update of the discriminator and the generated distribution with convergence guarantee. By framing different variants of GANs under the convex optimization framework, the corresponding training algorithms can all be improved by pushing the generated distribution along the optimal direction. Experiments on two synthetic datasets demonstrate that the proposed formulation can effectively avoid mode collapse. It also achieves competitive quantitative evaluation scores on two benchmark real-world image datasets.

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

# 7 APPENDIX

## 7.1 PROOF OF THEOREM 1

The proof of convergence for dual-driven algorithms can be found in (Bertsekas & Tsitsiklis, 1989, Chapter 3).

The primal-dual-driven algorithm for continuous time update has been studied in (Feijer & Paganini, 2010). Here, we show the convergence for the discrete-time case.

We choose a step size $\alpha(t)$ that satisfies

$$\alpha(t) > 0, \sum_{t=1}^{\infty} \alpha(t) = \infty, \sum_{t=1}^{\infty} \alpha^2(t) < \infty. \tag{24}$$

Let $\boldsymbol{z}(t) = [\boldsymbol{x}(t), \boldsymbol{\lambda}(t)]^T$ be a vector consisting of the primal and dual variables at the $t$-th iteration. The primal-dual-driven update can be expressed as:

$$\boldsymbol{z}(t+1) = \boldsymbol{z}(t) + \alpha(t)\boldsymbol{T}(t), \tag{25}$$

where

$$T(t) = \begin{bmatrix} \partial_{\boldsymbol{x}} L(\boldsymbol{x}(t), \boldsymbol{\lambda}(t)) \\ -\partial_{\boldsymbol{\lambda}} L(\boldsymbol{x}(t), \boldsymbol{\lambda}(t)) \end{bmatrix} = \begin{bmatrix} \partial f_0(\boldsymbol{x}(t)) + \sum_{i=1}^{\ell} \lambda_i \partial f_i(\boldsymbol{x}(t)) \\ -F(\boldsymbol{x}(t)) \end{bmatrix}, \tag{26}$$

and

$$\boldsymbol{F}(\boldsymbol{x}) = \begin{bmatrix} f_1(\boldsymbol{x}) \\ \vdots \\ f_\ell(\boldsymbol{x}) \end{bmatrix}. \tag{27}$$

Since the subgradient is bounded by assumption, there exists $M > 0$ such that $||\boldsymbol{T}(\cdot)||_2^2 < M$, where $||.||_2$ stands for the $L_2$ norm.

Let $\boldsymbol{x}^*$ be the unique saddle point and $\Phi$ be the set of saddle points of $\boldsymbol{\lambda}$. For any $\boldsymbol{\lambda}^* \in \Phi$, it satisfies

$$L(\boldsymbol{x}, \boldsymbol{\lambda}^*) \leq L(\boldsymbol{x}^*, \boldsymbol{\lambda}^*) \leq L(\boldsymbol{x}^*, \boldsymbol{\lambda}). \tag{28}$$

for all $\boldsymbol{x}$ and $\boldsymbol{\lambda}$.

For any saddle point $(\boldsymbol{x}^*, \boldsymbol{\lambda}^*)$, we have

$$||\boldsymbol{x}(t+1) - \boldsymbol{x}^*||_2^2 = ||\mathcal{P}_X[\boldsymbol{x}(t) + \alpha(t)\partial_{\boldsymbol{x}}L(\boldsymbol{x}(t), \boldsymbol{\lambda}(t))] - \boldsymbol{x}^*||_2^2 \tag{29}$$

$$\leq ||\boldsymbol{x}(t) + \alpha(t)\partial_{\boldsymbol{x}}L(\boldsymbol{x}(t), \boldsymbol{\lambda}(t)) - \boldsymbol{x}^*||_2^2 \tag{30}$$

$$= ||\boldsymbol{x}(t) - \boldsymbol{x}^*||_2^2 + 2\alpha(t)\partial_{\boldsymbol{x}}L(\boldsymbol{x}(t), \boldsymbol{\lambda}(t))(\boldsymbol{x}(t) - \boldsymbol{x}^*)$$
$$+ \alpha^2(t)||\partial_{\boldsymbol{x}}L(\boldsymbol{x}(t), \boldsymbol{\lambda}(t))||_2^2 \tag{31}$$

$$\leq ||\boldsymbol{x}(t) - \boldsymbol{x}^*||_2^2 + 2\alpha(t)\partial_{\boldsymbol{x}}L(\boldsymbol{x}(t), \boldsymbol{\lambda}(t))(\boldsymbol{x}(t) - \boldsymbol{x}^*) + \alpha^2(t)M \tag{32}$$

where (30) is due to the nonexpansive projection lemma for any $X$ that contains $\boldsymbol{x}^*$ (Bertsekas & Tsitsiklis, 1989, Chapter 3), and (32) is due to the assumption that the subgradients are upper bounded by $M$.

Similarly, we have

$$||\boldsymbol{\lambda}(t+1) - \boldsymbol{\lambda}^*||_2^2 \leq ||\boldsymbol{\lambda}(t) - \boldsymbol{\lambda}^*||_2^2 - 2\alpha(t)\boldsymbol{F}(\boldsymbol{x})^T(\boldsymbol{\lambda}(t) - \boldsymbol{\lambda}^*) + \alpha^2(t)M. \tag{33}$$

Let $\boldsymbol{\lambda}^*(t) = \arg\min_{\boldsymbol{\lambda}^* \in \Phi} ||\boldsymbol{\lambda}(t) - \boldsymbol{\lambda}^*||_2$. Define $\boldsymbol{z}^*(t) = [\boldsymbol{x}^*, \boldsymbol{\lambda}^*(t)]$. Since (32) and (33) hold for any $\boldsymbol{\lambda}^* \in \Phi$, we have

$$||\boldsymbol{z}(t+1) - \boldsymbol{z}^*(t+1)||_2^2 = ||\boldsymbol{x}(t+1) - \boldsymbol{x}^*||_2^2 + ||\boldsymbol{\lambda}(t+1) - \boldsymbol{\lambda}^*(t+1)||_2^2 \tag{34}$$

$$\leq ||\boldsymbol{x}(t+1) - \boldsymbol{x}^*||_2^2 + ||\boldsymbol{\lambda}(t+1) - \boldsymbol{\lambda}^*(t)||_2^2 \tag{35}$$

$$\leq ||\boldsymbol{z}(t) - \boldsymbol{z}^*(t)||_2^2 + 2\alpha(t)\boldsymbol{T}^T(t)(\boldsymbol{z}(t) - \boldsymbol{z}^*(t)) + 2\alpha^2(t)M. \tag{36}$$

Next we will show that $(\boldsymbol{x}(t), \boldsymbol{\lambda}(t))$ converges to a saddle point. The intuition is that for large $t$, the second term (36) is less than zero and dominates over the third term, thus $\boldsymbol{z}(t)$ will be driven to the set of saddle points.

Since $L(\boldsymbol{x}, \lambda)$ is concave in $\boldsymbol{x}$ and convex in $\boldsymbol{\lambda}$, we have

$$\partial_{\boldsymbol{x}}L(\boldsymbol{x}(t), \boldsymbol{\lambda}(t))(\boldsymbol{x}(t) - \boldsymbol{x}^*) \leq L(\boldsymbol{x}(t), \boldsymbol{\lambda}(t)) - L(\boldsymbol{x}^*, \boldsymbol{\lambda}(t)) \tag{37}$$

$$\partial_{\boldsymbol{\lambda}}L(\boldsymbol{x}(t), \boldsymbol{\lambda}(t))(\boldsymbol{\lambda}(t) - \boldsymbol{\lambda}^*) \geq L(\boldsymbol{x}(t), \boldsymbol{\lambda}(t)) - L(\boldsymbol{x}(t), \boldsymbol{\lambda}^*). \tag{38}$$

Therefore,

$$\boldsymbol{T}^T(t)(\boldsymbol{z}(t) - \boldsymbol{z}^*(t)) \leq L(\boldsymbol{x}(t), \boldsymbol{\lambda}(t)) - L(\boldsymbol{x}^*, \boldsymbol{\lambda}(t)) + L(\boldsymbol{x}(t), \boldsymbol{\lambda}^*(t)) - L(\boldsymbol{x}(t), \boldsymbol{\lambda}(t)) \tag{39}$$

$$= L(\boldsymbol{x}^*, \boldsymbol{\lambda}^*(t)) - L(\boldsymbol{x}^*, \boldsymbol{\lambda}(t)) + L(\boldsymbol{x}(t), \boldsymbol{\lambda}^*(t)) - L(\boldsymbol{x}^*, \boldsymbol{\lambda}^*(t)) \tag{40}$$

$$\leq 0, \tag{41}$$

where the last step is due to the definition of saddle point (28). Combining (36) and (41), we have

$$||\boldsymbol{z}(t+1) - \boldsymbol{z}^*(t+1)||_2^2 \leq ||\boldsymbol{z}(t) - \boldsymbol{z}^*(t)||_2^2 + 2\alpha^2(t)M. \tag{42}$$

Summing (42) over $1 \leq t \leq n - 1$, we have

$$||\boldsymbol{z}(n) - \boldsymbol{z}^*(n)||_2^2 \leq ||\boldsymbol{z}(1) - \boldsymbol{z}^*(1)||_2^2 + \sum_{t=1}^{n-1} 2\alpha^2(t)M. \tag{43}$$

Since the saddle points are bounded by assumption, the initial point $||\boldsymbol{z}(1)||_2$ is bounded and $\sum_{t=1}^{\infty} \alpha^2(t)$ is bounded, $||\boldsymbol{z}(n)||_2$ must be bounded.

Give any $\epsilon > 0$, define a neighbor of the saddle points as

$$A_\epsilon = \{\boldsymbol{z} : ||\boldsymbol{z} - (\boldsymbol{x}^*, \boldsymbol{\lambda}^*)||_2^2 < \epsilon, \exists \boldsymbol{\lambda}^* \in \Phi\}. \tag{44}$$

We first show that there must be infinitely many points of $\boldsymbol{z}(t)$ that are in $A_\epsilon$. Suppose this does not hold, then there exists some $N_0 > 0$, such that for every $t \geq N_0$, $\boldsymbol{z}_t \notin A_\epsilon$. By the continuity of

function $L(\cdot, \cdot)$, (40) implies that there exists some $\delta > 0$ such that for every $t \geq N_0$, $\boldsymbol{T}^T(t)(\boldsymbol{z}(t) - \boldsymbol{z}^*(t)) < -\delta$. In this case, by summing (36) over $N_0 \leq t \leq n-1$, we have

$$||\boldsymbol{z}(n) - \boldsymbol{z}^*(n)||_2^2 \leq ||\boldsymbol{z}(N_0) - \boldsymbol{z}^*(N_0)||_2^2 - 2\sum_{t=N_0}^{n-1} \alpha(t)\delta + \sum_{t=N_0}^{n-1} 2\alpha^2(t)M. \qquad (45)$$

Note that $||\boldsymbol{z}(N_0) - \boldsymbol{z}^*(N_0)||_2^2$ is bounded. By the choice of the step size (24), we have $||\boldsymbol{z}(n) - \boldsymbol{z}^*(n)||_2^2$ tends to $-\infty$, which is contradicted with the fact that $||\boldsymbol{z}(n) - \boldsymbol{z}^*(n)||_2^2 \geq 0$. Therefore, there are infinitely many $\boldsymbol{z}(t)$ in $A_\epsilon$.

Consequently, we can find a large enough $N_1$ such that $2M\sum_{t=N_1}^{\infty} \alpha^2(t) \leq \epsilon$ and $\|\boldsymbol{z}(N_1) - \boldsymbol{z}^*(N_1)\| \leq \epsilon$. Summing (42) over $N_1 \leq t \leq n-1$ we have

$$||\boldsymbol{z}(n) - \boldsymbol{z}^*(n)||_2^2 \leq ||\boldsymbol{z}(N_1) - \boldsymbol{z}^*(N_1)||_2^2 + 2M\sum_{t=N_1}^{n-1} \alpha^2(t) \qquad (46)$$

$$\leq 2\epsilon. \qquad (47)$$

In other words, for any $\epsilon > 0$, there exists $N_1 > 0$ such that for all $t \geq N_1$, $||\boldsymbol{z}(t) - \boldsymbol{z}^*(t)||_2 \leq 2\epsilon$. Since it holds for all $\epsilon$, it implies that there are infinitely many $\boldsymbol{z}(t)$ that belong to the saddle points.

The set of saddle points is compact by assumption. By the Bolzano–Weierstrass theorem, there must exist a subsequence of $\{\boldsymbol{z}(t_n)\}$ that converges to a saddle point $\boldsymbol{z}_0$. For such subsequence, there exists some large enough $n_0$ such that $t_{n_0} \geq N_1$ and $||\boldsymbol{z}(t_n) - \boldsymbol{z}_0||_2^2 \leq \epsilon$, for every $n \geq n_0$. Since (42) holds for any saddle point $\boldsymbol{z}^*(t)$, we replace $\boldsymbol{z}^*(t)$ by $\boldsymbol{z}_0$ and sum (42) over $t_{n_0} \leq t \leq n-1$ to obtain

$$||\boldsymbol{z}(n) - \boldsymbol{z}_0||_2^2 \leq ||\boldsymbol{z}(t_{n_0}) - \boldsymbol{z}_0||_2^2 + 2M\sum_{t=t_{n_0}}^{n-1} \alpha^2(t) \qquad (48)$$

$$\leq ||\boldsymbol{z}(t_{n_0}) - \boldsymbol{z}_0||_2^2 + 2M\sum_{t=N_1}^{n-1} \alpha^2(t) \qquad (49)$$

$$\leq 2\epsilon. \qquad (50)$$

This means that for any $\epsilon$, there exists $N_2 = t_{n_0}$ such that for every $t \geq N_2$, $||\boldsymbol{z}(t) - \boldsymbol{z}_0||_2^2 \leq \epsilon$. That concludes the proof that $\boldsymbol{z}(t)$ converges to a saddle point.

## 7.2 1000 CLASS MNIST DATASET

We use an augmented version of MNIST dataset similar to the experiment conducted in (Che et al., 2016; Metz et al., 2016). Each image in this dataset is created by randomly choosing three letter images from MNIST dataset. The three images are stacked as the R,G, and B channels into a color image. This dataset has 1000 distinct modes, corresponding to each combination of the ten MNIST classes in each channel.

We train a GAN and a classifier on this dataset. For each generated image, we apply the classifier to determine its label. We compute two metrics on this dataset. the number of modes the GAN generates, and the inception score computed using the classifier. We use the same architecture as (Metz et al., 2016) in our experiment. The result is shown in Table 7.2.

We find that the proposed method achieves a much better performance than unrolled GAN with 5 steps and comparable performance to unrolled GAN with 10 steps in terms of the number of predicted modes. However, since our method does not involve unrolling step, it is much more computationally efficient. Notice that although (Che et al., 2016) generates much more modes, it uses a more complex architecture, and such architecture is known to contribute to mode collapse avoidance on the 1000 Class MNIST dataset (Metz et al., 2016). Compared to the baseline that does not use the second regularization term in (13), the proposed method achieves better inception score, and it generates more modes.

Table 3: Performance comparison on augmented MNIST dataset.

| Method | Modes generated | Inception Score |
|---|---|---|
| (Metz et al., 2016) 5 steps | 732 | NA |
| (Metz et al., 2016) 10 steps | 817 | NA |
| (Che et al., 2016) | 969 | NA |
| Baseline | 526 | 87.15 |
| Proposed | 827 | 155.6 |

## 7.3 TOY EXAMPLE TRAINING DETAILS

In the toy example, both the generator and the discriminator has only one ReLU hidden layer with 64 neurons. The output activation is sigmoid function for GAN and ReLU for WGAN. For WGAN, the parameters are clipped in between $[-1, 1]$, and the networks are trained with Root Mean Square Propagation (RMSProp) with a learning rate of 1e-4. For GAN, the networks are trained with Adam with a learning rate of 1e-4. The minibatch size is 32. The bandwidth parameter for the Gaussian kernel is initialized to be $\sigma = 0.5$ and then is changed to 0.1 after 2000 iterations.

## 7.4 2D MIXTURE GAUSSIAN DATA TRAINING DETAILS

We use the network structure in (Metz et al., 2016) to evaluate the performance of our proposed method. The data is sampled from a mixture of 8 Gaussians of standard deviation of 0.02 uniformly located on a circle of radius 2. The noise samples are a vector of 256 independent and identically distributed (i.i.d.) Gaussian variables with mean zero and standard deviation of 1.

The generator has two hidden layers of size 128 with ReLU activation. The last layer is a linear projection to two dimensions. The discriminator has one hidden layer of size 128 with ReLU activation followed by a fully connected network to a sigmoid activation. All the biases are initialized to be zeros and the weights are initalilzed via the "Xavier" initialization (Glorot & Bengio, 2010). The training follows the primal-dual-driven algorithm, where both the generator and the discriminator are updated once at each iteration. The Adam optimizer is used to train the discriminator with 8e-4 learning rate and the generator with 4e-4 learning rate. The minibatch sample number is 64.

## 7.5 MNIST TRAINING DETAILS

For MNIST dataset, the generator network is a deconvolutional neural network. It has two fully connected layer with hidden size 1024 and $7 \times \times 7 \times 128$, two deconvolutional layers with number of units 64, 32, stride 2 and deconvolutional kernel size $4 \times 4$ for each layer, respectively, and a final convolutional layer with number of hidden unit 1 and convolutional kernel $4 \times 4$.. The discriminator network is a two layer convolutional neural network with number of units 64, 32 followed by two fully connected layer of hidden size 1024 and 1. The input noise dimension is 64.

We employ ADAM optimization algorithm with initial learning rate 0.01 and $\beta = 0.5$.

## 7.6 CIFAR TRAING DETAILS

For CIFAR dataset, the generator is a 4 layer deconvolutional neural network, and the discriminator is a 4 layer convolutional neural network. The number of units for discriminator is $[64, 128, 256, 1024]$, and the number of units for generator is $[1024, 256, 128, 64]$. The stride for each deconvolutional and convolutional layer is two.

We employ RMSProp optimization algorithm with initial learning rate of 0.0001, decay rate 0.95, and momentum 0.1.

