# OpenReview forum: "TRAINING GENERATIVE ADVERSARIAL NETWORKS VIA PRIMAL-DUAL SUBGRADIENT METHODS: A LAGRANGIAN PERSPECTIVE ON GAN"
_ICLR.cc/2018/Conference — Accept (Poster)_

### Official Review · AnonReviewer1 · 2017-11-25
**Interesting insights although gap between theory and practice should be discussed**

**Rating:** 7
**Confidence:** 4

**Review:**

This paper formulates GAN as a Lagrangian of a primal convex constrained optimization problem. They then suggest to modify the updates used in the standard GAN training to be similar to the primal-dual updates typically used by primal-dual subgradient methods.

Technically, the paper is sound. It mostly leverages the existing literature on primal-dual subgradient methods to modify the GAN training procedure. I think this is a nice contribution that does yield to some interesting insights. However I do have some concerns about the way the paper is currently written and I find some claims misleading.

Prior convergence proofs: I think the way the paper is currently written is misleading. The authors quote the paper from Ian Goodfellow: “For GANs, there is no theoretical prediction as to
whether simultaneous gradient descent should converge or not.”. However, the f-GAN paper gave a proof of convergence, see Theorem 2 here: https://arxiv.org/pdf/1606.00709.pdf. A recent NIPS paper by (Nagarajan and Kolter, 2017) also study the convergence properties of simultaneous gradient descent. Another problem is of course the assumptions required for the proof that typically don’t hold in practice (see comment below).

Convex-concave assumption: In practice the GAN objective is optimized over the parameters of the neural network rather than the generative distribution. This typically yields a non-convex non-concave optimization problem. This should be mentioned in the paper and I would like to see a discussion concerning the gap between the theory and the practical algorithm.

Relation to existing regularization techniques: Combining Equations 11 and 13, the second terms acts as a regularizer that minimizes [\lapha f_1(D(x_i))]^2. This looks rather similar to some of the recent regularization techniques such as
Improved Training of Wasserstein GANs, https://arxiv.org/pdf/1704.00028.pdf
Stabilizing Training of Generative Adversarial Networks through Regularization, https://arxiv.org/pdf/1705.09367.pdf
Can the authors comment on this? I think this would also shed some light as to why this approach alleviates the problem of mode collapse.

Curse of dimensionality: Nonparametric density estimators such as the KDE technique used in this paper suffer from the well-known curse of dimensionality. For the synthetic data, the empirical evidence seem to indicate that the technique proposed by the authors does work but I’m not sure the empirical evidence provided for the MNIST and CIFAR-10 datasets is sufficient to judge whether or not the method does help with mode collapse. The inception score fails to capture this property. Could the authors explore other quantitative measure? Have you considered trying your approach on the augmented version of the MNIST dataset used in Metz et al. (2016) and Che et al. (2016)?

Experiments
Typo: Should say “The data distribution is p_d(x) = 1{x=1}”.

---

> ### Author Response · Authors · 2018-01-02
> **Response to AnonReviewer1: More discussions and experiments have been added.**
>
> The authors would like to thank the reviewer for his/her invaluable comments. We have taken the reviewers' comments into consideration when revising our paper. Moreover, our responses to the comments raised by the reviewer are as follows:
>
> 1. Comment: Prior convergence proofs: I think the way the paper is currently written is misleading. The authors quote the paper from Ian Goodfellow: “For GANs, there is no theoretical prediction as to
> whether simultaneous gradient descent should converge or not.”. However, the f-GAN paper gave a proof of convergence, see Theorem 2 here: https://arxiv.org/pdf/1606.00709.pdf. A recent NIPS paper by (Nagarajan and Kolter, 2017) also study the convergence properties of simultaneous gradient descent. Another problem is of course the assumptions required for the proof that typically don’t hold in practice (see comment below).
>
> Our reply: We would like to thank the reviewer for pointing out the latest NIPS paper by Nagarajan and Kolter. We have included this in our literature review. We have also made revisions in the paper to avoid the misleading arguments (see below).
>
> \textit{ However, the analysis of the convergence properties on the training
> approaches is challenging, as noted by Ian Goodfellow in (Goodfellow, 2016), ``For GANs, there is
> no theoretical prediction as to whether simultaneous gradient descent should converge or not. Settling
> this theoretical question, and developing algorithms guaranteed to converge, remain important open
> research problems.". There have been some recent studies on the convergence behaviours of GAN
> training (Nowozin et al., 2016; Li et al., 2017; Heusel et al., 2017; Nagarajan \& Kolter, 2017;
> Mescheder et al., 2017). The simultaneous gradient descent method was proved to converge assuming
> the objective function is convex-concave in the network parameters (Nowozin et al., 2016). The local
> stability property is established in (Heusel et al., 2017; Nagarajan \& Kolter, 2017). }
>
> Ian Goodfellow raised the convergence issue in the tutorial paper, because the study of convergence for the simultaneous gradient descent method was limited at that time. They also gave a counterexample in the tutorial paper, which shows that the simultaneous gradient descent cannot converge for some objective functions with some step size. This is one of the motivations of the paper to study the simultaneous gradient descent method.
>
> We agree with the reviewer that this convergence issue has been studied at least in the following works:
> [a] ``f-GAN: Training Generative Neural Samplers using Variational Divergence Minimization'' by Nowozin and Cseke.
> [b] ``Gradient descent GAN optimization is locally stable" by Nagarajan and Kolter.
> [c] ``GANs trained by a two time-scale update rule converge to a Nash equibrium" by Heusel et al (as noted in our introduction part).
>
> These papers together with our paper study the convergence issue from different perspectives. In particular,  these three papers study the convergence behavior of updates over the network parameters.  Reference [a] assumes the objective function is convex-concave in the network parameters, while reference [b] and [c] study the local stability property. Our paper studies the convergence behavior in the function space, which was the started point in the first GAN paper by Ian Goodfellow. We incorporate the two conventional training methods into one framework, namely simultaneous gradient descent update and discrminator-driven update (discriminator is fully optimized before the gradient update of the generator). The theoretical convergence proof leverage some well-established results from the primal-dual subgradient methods for convex optimization. Although the actual optimization is over the network parameters, which is non-convex non-cave in general, our formulation provides important insights in improving the training methods, as detailed in the next point.

---

> > ### Author Response · Authors · 2018-01-02
> > **Reply to comment 2.**
> >
> > 2. Comment: Convex-concave assumption: In practice the GAN objective is optimized over the parameters of the neural network rather than the generative distribution. This typically yields a non-convex non-concave optimization problem. This should be mentioned in the paper and I would like to see a discussion concerning the gap between the theory and the practical algorithm.
> >
> > Our reply: We agree with the reviewer that the actual optimization is over the network parameters, which is non-convex non-concave in general. We mentioned this important point in the last paragraph of Section 3.
> >
> > Although the local stability property is studied in Nagarajan 2017 and Heusel 2017, the convergence to the global optimum point is not guaranteed. Even for non-adversarial training, the non-convex is not well-understood. In fact, a non-convex optimization is NP hard in general.
> >
> > To complement the gap between theory and practical algorithm, most of the works including the first GAN paper by Ian Goodfellow, the f-GAN paper and the recent paper ``Training GANs with Optimism" apply the following approach. First, they propose methods that have good theoretical properties in the convex setting. Secondly, the methods are applied in the non-convex setting, and the actual learning performances are evaluated by experiments, which hopefully yield promising results. Our paper follows this approach as well.
> >
> > In this paper, we build the connection between GAN training and finding the saddle points of the Lagrangian function for a convex optimization problem. Although the theoretical convergence proof assumes functional space updates, the relationship provides important insights in understanding GANs and designing training methods. It inspires an improved training technique to avoid mode collapse by pushing the updates of the generated probabilities along the optimal direction in function space.
> >
> > For example, when mode collapse occurs, the generated probability at some data point $x$ is zero or very small and the discriminator output $D(x)$ is close to 1. Using the traditional training, the loss function at point $x$ contributes little since the derivative at that point is almost zero. We know that the ideal update direction in the function space according to the primal-dual update rule is given by Eq. (11), which gives a large gradient to push the generator to produce some samples at $x$. The synthetic example shows that it indeed increases the data sample diversity and effectively avoids mode collapse, which may never be escaped by the conventional GAN and WGAN.

---

> > > ### Author Response · Authors · 2018-01-02
> > > **Reply to comment 3**
> > >
> > >
> > > 3. Comment: Relation to existing regularization techniques: Combining Equations 11 and 13, the second terms acts as a regularizer that minimizes $[\alpha f_1(D(x_i))]^2$. This looks rather similar to some of the recent regularization techniques such as
> > >
> > > [a] Improved Training of Wasserstein GANs, https://arxiv.org/pdf/1704.00028.pdf
> > > [b] Stabilizing Training of Generative Adversarial Networks through Regularization, https://arxiv.org/pdf/1705.09367.pdf
> > >
> > > Can the authors comment on this? I think this would also shed some light as to why this approach alleviates the problem of mode collapse.
> > >
> > > Our reply:  We would like to thank the reviewer for this inisightful question. We would like to point out the differences between our regularization term and other works. We also incorporate the discussions in the revised paper.
> > >
> > > The regularization terms proposed in different papers may have different purposes:
> > > (1).  In reference (a), the gradient penalty regularization is calculated as
> > > $$(\nabla_{\hat{x}} D(\hat{x}) -1)^2,$$
> > > where $\hat{x}$ is some point lying in between the data samples and the generated samples. It is used to force the gradient norm to be close to 1, in order to enforce the Lipschitz constraint of the discriminator function. The recent paper ``Spectral Normalization for Generative Adversarial Networks" also aims to regularize the gradient norm.
> > >
> > > (2). The regularization term in reference (b) is calculated as
> > > $$E_{p_{g}}[f^{c''}  \circ \psi ||\nabla \psi||^2],$$
> > > where $f^c(\cdot)$ is the Fenchel dual of the f-divergence and $\psi$ is the discriminator function. It is used to smooth the probability distribution such that the generated distribution is not disjoint with the data distribution. In particular, the regularization term was shown to have the same effects of adding noise perturbation in the discriminator input, as suggested by Martin Arjovsky and Léon Bottou in ``Towards principled methods for training generative adversarial networks".
> > >
> > >
> > > (3). The regularization term in ``"Mode Regularized Generative Adversarial Networks" by Che et al is calculated as
> > > $$||\bx_i - G(E(x_i))||^2,$$
> > > where $E(\cdot)$ is the autoencoder for the data samples. The regularization term is used to penalize the missing modes by minimizing the Euclidean distance between the data samples and the generated samples.
> > >
> > >
> > > The purpose of the regularization term in our paper is more aligned with the third one, with the aim of avoiding missing modes.  However, as discussed in the introduction part in the paper, whether the regularization in (Che et al, 2016) is optimal or whether it can converge lacks theoretical guarantee. In this paper, we leverage the insights of the primal-dual subgradient methods to force the generated distributions to update according to the optimal direction:
> > > $$||p'_g(\bx_i) -  \sum_j \frac{1}{m_2} \sum_{j=1}^{m_2} k_{\sigma} (G(\bz_j)-\bx_i) ||.$$  (Eq.(11) in the paper)
> > >
> > > Note that the regularization term in our paper is actually not $[\alpha f_1(D(x_i))]^2$. The reviewer has this confusion probably because Eq. (11) is directly substituted in Eq. (13). We have modified the notation in the draft to avoid such confusion. The proposed regularization is motivated due to the following reasoning.
> > >
> > > By the primal-dual subgradient method, we know that the updated generated distribution $p'_g$ should be updated according to Eq. (11). Then we fix the target probability distribution $p'_g$ and optimize the generator such that its generated distribution approximated by Eq. (12) is pushed to $p'_g$. As discussed in Section 3.3, when a missing mode occurs at some $x_i$, the generated probability at $x_i$ is close to 0 and $D(x_i)$ is close to 1. Then the term $\alpha f_1(D(x_i)) = \alpha \log ( 2 (1-D(x_i)))$ would be very large and the regularization term plays an important role in the loss function. Ideally, it should encourage the generator to generate some samples at $x_i$, because the loss function would be very large otherwise. For every data point $x_i$, we are using the information from all the data in a batch in the regularization term while only the data point $x_i$ itself is used in (Che et al, 2016). The formulation also enables us to derive different kinds of regularizers for different GAN variants.

---

> > > > ### Author Response · Authors · 2018-01-02
> > > > **Reply to other comments**
> > > >
> > > > 4. Comment: Curse of dimensionality: Nonparametric density estimators such as the KDE technique used in this paper suffer from the well-known curse of dimensionality. For the synthetic data, the empirical evidence seem to indicate that the technique proposed by the authors does work but I’m not sure the empirical evidence provided for the MNIST and CIFAR-10 datasets is sufficient to judge whether or not the method does help with mode collapse. The inception score fails to capture this property. Could the authors explore other quantitative measure? Have you considered trying your approach on the augmented version of the MNIST dataset used in Metz et al. (2016) and Che et al. (2016)?
> > > >
> > > > Our reply: We agree with the reviewer that the curse of dimension is a known problem for nonparametric density estimation. We would also like to acknowledge some recent works that show promising results in generative learning using nonparametric density estimation, including
> > > >
> > > > [a]  "``Generative Moment Matching Networks" https://arxiv.org/abs/1502.02761,
> > > > [b] ``"MMD GAN: Towards Deeper Understanding of Moment Matching Network", https://arxiv.org/abs/1705.08584
> > > >
> > > > [c] ``"Non-parametric estimation of Jensen-Shannon Divergence in Generative Adversarial Network training", https://arxiv.org/pdf/1705.09199.pdf.
> > > >
> > > > In practice, in order to work with high dimensional data such as large images, it is useful to project data into lower dimension with pretrained neural network. Popular choice of projections include the bottleneck layer of a classifier neural network trained on ImageNet and auto-encoder (used in Che et al. 2016). This approach has also been used in references [b] and [c], and show excellent performance for large datasets.
> > > >
> > > >
> > > > We can incorporate the projection neural network and apply the KDE on the projected space $f_{\phi} (\mathcal{X})$ but not directly on the original data $\mathcal{X}$. The estimated probabilies become
> > > > \begin{align}
> > > > p_g (\bx_i) = \frac{1}{m_2} \sum_{j=1}^{m_2} k_{\sigma} (f_{\phi} (G(\bz_j))- f_{\phi} (\bx_i)).
> > > > \end{align}
> > > > We will leave it as our future work.
> > > >
> > > > We have experimented the proposed method on the augmented MNIST dataset with 1000 classes as proposed in unrolled GAN (Metz et al. 2016). The results are shown in the table below and details are elaborated in the appendix section.
> > > >
> > > > Method                                   & Modes generated & Inception Score
> > > >   Metz et al. (2016) 5 steps   & 732                          & NA
> > > >   Metz et al. (2016) 10 steps & 817                          & NA
> > > >   Che et al. (2016)                  & 969                          & NA
> > > >   Baseline                               & 526                           & 87.15
> > > >   Proposed                             & 827                           & 155.6
> > > >
> > > > We use the same architecture as unrolled GAN in the experiment. We find that the proposed approach generates much larger number of modes than unrolled GAN with 5 steps, and similar number of modes compared to unroll GAN with 10 steps. However, the proposed approach is much more computationally efficient than unrolled GAN. We also show that the proposed approach generates more modes and achieves higher inception score than the baseline, which does not use the regularization term in the modified training objective function of Eq. (13). For Che et al. (2016), it only misses 31.6 modes, but it uses a much more complex neural network architecture, which is known to contribute to mode collapse avoidance, as noted in (Metz et al. 2016).
> > > >
> > > > 5. Comment: Typo: Should say ``The data distribution is "$p_d(x) = 1\{x=1\}$”.
> > > >
> > > > Our reply: Corrected accordingly. We appreciate that the reviewer points this out.

---

> > > > > ### Comment · AnonReviewer1 · 2018-01-02
> > > > > **Please update the paper accordingly**
> > > > >
> > > > > I think the practical issue of performing KDE in high-dimensional spaces is not discussed enough in the revised version of the paper. Please revise the paper accordingly, clearly pointing the potential shortcomings and citing some of the work you just discussed here.
> > > > >
> > > > > This is still my main concern about the practical aspect of the proposed approach and I will therefore not raise my scores unless the authors can provide convincing experiments.

---

> > > > > > ### Author Response · Authors · 2018-01-03
> > > > > > **Updated paper accordingly**
> > > > > >
> > > > > > We would like to thank the reviewer for raising this up. We added the discussion in Section 3.2 of the paper (see below).
> > > > > >
> > > > > > \textit{Although having good convergence guarantee in theory, the non-parametric kernel density estimation
> > > > > > of the generated distribution may suffer from the curse of dimension. Previous works combining
> > > > > > kernel learning and the GAN framework have proposed methods to scale the algorithms to deal with
> > > > > > high-dimensional data, and the performances are promising (Li et al., 2015; 2017a; Sinn \& Rawat,
> > > > > > 2017). One common method is to project the data onto a low dimensional space using an autoencoder or a bottleneck layer of a pretrained neurual network, and then apply the kernel-based estimates on the feature space. Using this approach, the estimated probability of $\bx_i$ becomes
> > > > > > \begin{align}\label{eq:calc_prob}
> > > > > > p_g (\bx_i) = \frac{1}{m_2} \sum_{j=1}^{m_2} k_{\sigma} (f_{\phi} (G(\bz_j) )-f_{\phi}( \bx_i )),
> > > > > > \end{align}
> > > > > > where $f_{\phi} (.)$ is the projection of the data to a low dimensional space. We will leave the work of generating high-resolution images using this approach as future work. }
> > > > > >
> > > > > > We agree with the reviewer that it will definitely help with more convincing experiments. Since the application of kernel methods with data projected onto a low-dimensional space has shown promising results in the previous works, we think it should be applicable to our approach as well. Nevertheless, training a GAN for high-dimensional data such as high resolution images may require more sophisticated GAN architectures, similar to the paper of "``Progressive Growing of GANs for Improved Quality, Stability, and Variation''. We will leave this as our future work.

---

> > > > ### Comment · AnonReviewer1 · 2018-01-02
> > > > **Clarification needed**
> > > >
> > > > Thanks for the detailed answer. Can you confirm that all the regularization methods cited above would avoid the mode collapse problem for the specific example you used in your paper?

---

> > > > > ### Author Response · Authors · 2018-01-03
> > > > > **Clarify on the regularization methods**
> > > > >
> > > > > Comment:  Can you confirm that all the regularization methods cited above would avoid the mode collapse problem for the specific example you used in your paper?
> > > > >
> > > > > Our reply: Since we did not implement all the methods, we can only provide some arguments from the perspective of theory.
> > > > >
> > > > > [a] Improved Training of Wasserstein GANs.
> > > > > [b] Stabilizing Training of Generative Adversarial Networks through Regularization.
> > > > > [c] Mode Regularized Generative Adversarial Networks.
> > > > >
> > > > > For WGAN, the purpose of weight clipping and gradient norm penalty in reference [a] is to enforce the Lipschitz condition of the discriminator function. The toy example shows that even with optimization over the functions with Lipschitz constraints, mode collapse still occurs.
> > > > >
> > > > > For reference [b], the regularization technique has the effects of adding noise purturbation in the discriminator input, so that the support of data plus noise will be overlapped with the support of generated data plus noise. Without noise addition, the true data support may be disjoint with the generated data, which yields mode collapse. If the support of noise is large enough, the noise perturbation technique should be able to alleviate the mode collapse problem. However, according to our knowledge, the regularizer applied in reference [b] is derived from first-order Taylor expansion, assuming the perturbation is {\em small}. Therefore, it may require a fine tuning of the noise variance (equivalent to the weighting factor in the regularizer) in practical training, especially in high dimensional datasets.
> > > > >
> > > > > For reference [c], assume $E(.)$ can effectively encode the data to the latent space, the regularization term should be effective to encourage the generator to generate samples in all modes. Thus, it should be able to alleviate mode collapse problem, and the authors of [c] did some experiments to demonstrate the performance  in their paper.

---

> > > > > > ### Comment · AnonReviewer1 · 2018-01-04
> > > > > > **Thanks for clarifying this point.**
> > > > > >
> > > > > > Thanks for clarifying this point.

---

> > ### Comment · AnonReviewer1 · 2018-01-02
> > **Thank you for the updated discussion**
> >
> > Thank you for the updated discussion. I think the new version nicely reflects the current status of existing convergence results for GAN training.

---

### Official Review · AnonReviewer3 · 2017-11-30
**Clarity analysis, very good motivation, but relatively limited novelty**

**Rating:** 6
**Confidence:** 4

**Review:**

This paper proposed a framework to connect the solving of GAN with finding the saddle point of a minimax problem.
As a result, the primal-dual subgradient methods can be directly introduced to calculate the saddle point.
Additionally, this idea not only fill the relatviely lacking of theoretical results for GAN or WGAN, but also provide a new perspective to modify the GAN-type models.
But this saddle point model reformulation  in section 2 is quite standard, with limited theoretical analysis in Theorem 1.
As follows, the resulting algorithm 1 is also standard primal-dual method for a saddle point problem.
Most important I think, the advantage of considering GAN-type model as a saddle point model is that first--order methods can be designed to solve it. But the numerical experiments part seems to be a bit weak, because the MINST or CIFAR-10 dataset is not large enough to test the extensibility for large-scale cases.

---

> ### Author Response · Authors · 2018-01-02
> **Response to AnonReviewer3**
>
> The authors would like to thank the reviewer for his/her invaluable comments. We have taken the reviewers' comments into consideration when revising our paper. Moreover, our responses to the comments raised by the reviewer are as follows:
>
> 1. Comment: But this saddle point model reformulation in section 2 is quite standard, with limited theoretical analysis in Theorem 1. As follows, the resulting algorithm 1 is also standard primal-dual method for a saddle point problem. Most important I think, the advantage of considering GAN-type model as a saddle point model is that first order methods can be designed to solve it.
>
> Our reply: We agree with the reviewer that the Lagrangian formulation in Section 2 is standard. The main contribution of the paper is to provide a new perspective of understanding GAN. In particular, we relate the minimax game to finding the saddle points of the Lagrangian function for a convex optimization problem, where the generated distribution plays the role of the dual variable.
>
> This inherent connection was not established in previous works and it shows that the standard training of GANs actually falls in the framework of primal-dual subgradient methods for convex optimization. As the the reviewer mentions, one important result is to show that the training actually converge to the optimal point if a proper step size is chosen, and both the discriminator output and the generated distribution are correctly updated according to the primal-dual rule. Besides this, it provides the following important insights:
>
> (a). It inspires an improved training technique to avoid mode collapse. In practical training, the generated distribution is not updated according to the desired direction. As Claim 1 points out, when the generated probability at some data point $\bx$ is zero and the discriminator output $D(\bx)$ is locally constant, mode collapse occurs. Using the traditional training, we can hardly avoid such mode collapse, even under the recently proposed WGAN. The Lagrangian formulation tells that the optimal update direction of $p_g(\cdot)$ is given by Eq. (11). When mode collapse occurs, Eq. (13) gives a large gradient to push the generator to produce some samples at $\bx$. The synthetic example shows that it indeed increases the data sample diversity and effectively avoids mode collapse.
>
> (b). It naturally incorporates different variants of GANs into the convex optimization framework including the
> family of f-GAN (Nowozin et al., 2016) and an approximate variant of WGAN. For all these GAN variants, an improved training objective can be easily derived.
>
> (c). The simultaneous primal-dual update is known to have a very slow convergence rate. There have been proposed methods to accelerate the convergence rates in the following papers:
>
> Angelia Nedic and Asuman Ozdaglar, ``Subgradient methods for saddle-point problems".
>
> Yunmei Chen, Guanghui Lan and Yuyuan Ouyang, ``Optimal primal-dual methods for a class of saddle point problems".
>
> By building the relation of GAN training and the primal-dual approach for convex optimizations, these improved methods can be directly applied.  In future research, we will evaluate the acceleration of the training process using these approaches.
>
> (d). For some GAN variants, where the objective function is not strictly convex, the convergence may be slow or the converging point is not unique. By casting the minimax game in the Lagrangian framework, we could easily tweak the objective function such that the objective function is strictly convex and the optimal solution is not affected, then the convergence performance can be improved. Examples can be found in ``Nonlinear Programming" by D. Bertsekas.

---

> > ### Author Response · Authors · 2018-01-02
> > **Reply to comment 2: more experiments are added.**
> >
> > 2. Comment: But the numerical experiments part seems to be a bit weak, because the MINST or CIFAR-10 dataset is not large enough to test the extensibility for large-scale cases.
> >
> > Our reply: We appreciate that the reviewer points this out. We have experimented with augmented MNIST dataset with 1000 classes as proposed in unrolled GAN (Metz et al. 2016). The results are shown in the table below and the details are elaborated in the appendix section.
> >
> >    Method                                         Modes generated     Inception Score
> >   Metz et al. (2016) 5 steps           732                               NA
> >   Metz et al. (2016) 10 steps         817                               NA
> >   Che et al. (2016)                          969                               NA
> >   Baseline                                        526                               87.15
> >   Proposed                                      827                               155.6
> >
> > We use the same architecture as unrolled GAN in the experiment. We find that the proposed approach generates much larger number of modes than unrolled GAN with 5 steps, and similar number of modes compared to unrolled GAN with 10 steps. However, the proposed approach is much more computationally efficient than unrolled GAN. We also show that the proposed approach generates more modes and achieves higher inception score than the baseline, which does not use the regularization term in the modified training objective function of Eq. (13). For Che et al. (2016), it only misses 31.6 modes, but it uses a much more complex neural network architecture, which is known to contribute to mode collapse avoidance, as noted in (Metz et al. 2016).

---

### Official Review · AnonReviewer2 · 2017-12-02
**Important problem with well evaluated solution.**

**Rating:** 7
**Confidence:** 3

**Review:**

In this paper, the authors study the relationship between training GANs and primal-dual subgradient methods for convex optimization. Their technique can be applied on top of existing GANs and can address issues such as mode collapse. The authors also derive a GAN variant similar to WGAN which is called the Approximate WGAN. Experiments on synthetic datasets demonstrate that the proposed formulation can avoid mode collapse. This is a strong contribution

In Table 2 the difference between inception scores for DCGAN and this approach seems significant to ignore. The authors should explain more possibly.
There is a typo in Page 2 – For all these varaints, -variants.

---

> ### Author Response · Authors · 2018-01-02
> **Response to AnonReviewer2**
>
> The authors would like to thank the reviewer for his/her invaluable comments. We have taken the reviewers' comments into consideration when revising our paper. Moreover, our responses to the comments raised by the reviewer are as follows:
>
> 1. Comment: In Table 2 the difference between inception scores for DCGAN and this approach seems significant to ignore. The authors should explain more possibly.
>
> Our reply:  The different performance is in part due to the different network architecture and different training objective  from DCGAN. Specifically,
> - We do not use BatchNorm.
> - We do not use LeakyReLU activation in the discriminator.
> - We do  not use SoftPlus for the last layer of discriminator.
> - We use the approximate-WGAN variant as proposed in the paper, while DCGAN uses the vanilla GAN objective function.
>
> In this regard, a more suitable baseline approach to compare is probably the WGAN result, which has similar architecture and optimization objective. In order to achieve better inception score performance, we probably need more extensive hyper-parameter tuning. We have to acknowledge that the aim of this paper is not to achieve superior performance, but to provide a new perspective on understanding GAN, and provide a new training technique that can be applied on top of different GAN variants to alleviate the mode collapse issue.
>
> 2. Comment: There is a typo in Page 2 – For all these varaints, -variants.
>
> Our reply: Corrected accordingly. We appreciate that the reviewer points out this mistake.

---

### Author Response · Authors · 2018-01-02
**To all reviewers: Thank you for the insightful comments!**

The authors would like to thank the reviewers for their insightful comments. We have taken the reviewers' comments into consideration when revising our paper. In particular, we have made the following major revisions:

1. We have corrected the typos and grammar mistakes as pointed out by the reviewers.

2. We have incorporated more references as pointed out by the reviewer in the literature survey. Moreover, we made some revisions in the discussions to better clarify our ideas.

3. We have run more experiments on the augmented MNIST dataset with 1000 classes to test the extensibility of our method for large-scale cases. Due to the page limits by the submission guideline, we elaborate the results in the appendix section.

---

### Decision · Program_Chairs · 2018-01-29
**ICLR 2018 Conference Acceptance Decision**

**Decision:**

Accept (Poster)

**Comment:**

The paper makes a good theoretical contribution by formulating the GAN training as primal-dual subgradient method for convex optimization and providing convergence proof. The authors then propose a modified objective to standard GAN training, based on this formulation, that helps address the mode collapse issue.
One weak point of the paper as pointed out by reviewers is that that the experimental results are underwhelming and the approach may not scale well to high dimensional datasets / high-resolution images. Interestingly, the proposed approach is general enough to be applied to other GAN variants that may address this issue in future. I recommend acceptance.